# Moonshine: Distilling with Cheap Convolutions

**Elliot J. Crowley**
School of Informatics
University of Edinburgh
elliot.j.crowley@ed.ac.uk

**Gavin Gray**
School of Informatics
University of Edinburgh
g.d.b.gray@ed.ac.uk

**Amos Storkey**
School of Informatics
University of Edinburgh
a.storkey@ed.ac.uk

## Abstract

Many engineers wish to deploy modern neural networks in memory-limited settings; but the development of flexible methods for reducing memory use is in its infancy, and there is little knowledge of the resulting cost-benefit. We propose structural model distillation for memory reduction using a strategy that produces a *student* architecture that is a simple transformation of the *teacher* architecture: no redesign is needed, and the same hyperparameters can be used. Using attention transfer, we provide Pareto curves/tables for distillation of residual networks with four benchmark datasets, indicating the memory versus accuracy payoff. We show that substantial memory savings are possible with very little loss of accuracy, and confirm that distillation provides student network performance that is better than training that student architecture directly on data.

## 1 Introduction

Despite advances in deep learning for a variety of tasks (LeCun et al., 2015), deployment of deep learning into embedded devices e.g. wearable devices, digital cameras, vehicle navigation systems, has been relatively slow due to resource constraints under which these devices operate. Big, memory-intensive neural networks do not fit on these devices, but do these networks have to be big and expensive? The dominant run-time memory cost of neural networks is the number of parameters that need to be stored. Can we have networks with substantially fewer parameters, without the commensurate loss of performance?

It is possible to take a large pre-trained *teacher* network, and use its outputs to aid in the training of a smaller *student* network (Ba & Caruana, 2014) through some distillation process. By doing this the student network is more powerful than if it was trained solely on the training data, and is closer in performance to the larger teacher. The lower-parameter student network typically has an architecture that is more shallow, or thinner — by which we mean its filters have fewer channels (Romero et al., 2015) — than the teacher. While it is not possible to arbitrarily approximate any network with another (Urban et al., 2017), the limit in neural network performance is at least in part due to the training algorithm, rather than its representational power.

In this paper, we take an alternative approach in designing our student networks. Instead of making networks thinner, or more shallow, we take the standard convolutional block such networks possess and replace it with a *cheaper* convolutional block, keeping the original architecture. For example, in a Residual Network (ResNet) (He et al., 2016a) this standard block is a pair of sequential $3\times3$ convolutions. We show that for a comparable number of parameters, student networks that retain the architecture of their teacher but with cheaper convolutional blocks outperform student networks with the original blocks and smaller architectures.

The cheap convolutional blocks we suggest are described in Section 3 as well as an overview of the methods we employ for distillation. In Section 4 we evaluate student networks with these blocks on CIFAR-10 and CIFAR-100 (Krizhevsky, 2009). Finally, in Section 5 we examine the efficacy of such networks for the tasks of ImageNet (Russakovsky et al., 2015) classification, and semantic segmentation on the Cityscapes dataset (Cordts et al., 2016). Our claims are as follows:

- Greater model compression by distillation is possible by replacing convolutional blocks than by shrinking the architecture.
- Grouped convolutional blocks, with or without a bottleneck contraction, are an effective replacement block.
- This replacement is cheap in design time (substitution), and cheap in training complexity; it uses the same optimiser and hyperparameters as those used during the original training.

## 2    Related Work

The parameters in deep networks have a great deal of redundancy; it has been shown that many of them can be predicted from a subset of parameters (Denil et al., 2013). However the challenge remains to find good ways to exploit this redundancy without losing substantial model accuracy. This observation, along with a desire for efficiency improvements has driven the development of smaller, and less computationally-intensive convolutions. One of the most prominent examples is the depthwise separable convolution (Sifre, 2014) which applies a separate convolutional kernel to each channel, followed by a pointwise convolution over all channels; depthwise separable convolutions have been used in several architectures (Ioffe & Szegedy, 2015; Chollet, 2016; Xie et al., 2017), and were explicitly adapted to mobile devices in Howard et al. (2017).

The depthwise part of this convolution is a specific case of a grouped convolution where there are as many groups as channels; grouped convolutions were used with a grouping of 2 (i.e. half the channels belong to each group) in the original AlexNet (Krizhevsky et al., 2012) due to GPU memory constraints. In the work of Ioannou et al. (2017) the authors examine networks trained from scratch for different groupings, motivated by efficiency. They found that these networks actually generalised better than an ungrouped alternative. However, separating the spatial and channel-wise elements is not the only way to simplify a convolution. In Jin et al. (2015) the authors propose breaking up the general 3D convolution into a set of 3 pointwise convolutions along different axes. Wang et al. (2016) start with separable convolutions and add topological subdivisioning, a way to treat sections of tensors separately, and a bottleneck of the spatial dimensions. Both of these methods produce models that are several times smaller than the original model while maintaining accuracy.

In a separable convolution, the most expensive part is the pointwise convolution, so it has been proposed that this operation could also be grouped over sets of channels. However, to maintain some connections between channels, it is helpful to add an operation mixing the channels together (Zhang et al., 2018). More simply, a squared reduction can be achieved by applying a bottleneck on the channels before the spatial convolution (Iandola et al., 2016; Xie et al., 2017). In this paper we examine the potency of a separable bottleneck structure.

The work discussed thus far involves learning a compressed network from scratch. There are alternatives to this such as retraining after reducing the number of parameters (Han et al., 2016; Li et al., 2017). We are interested in learning our smaller network as a student through distillation (Buciluǎ et al., 2006; Ba & Caruana, 2014) in conjunction with a pre-trained teacher network.

How small can our student network be? The complex function of a large, deep teacher network can, theoretically, be approximated by a network with a single hidden layer with enough units (Cybenko, 1989). The difficulty in practice is *learning that function*. Knowledge distillation (Ba & Caruana, 2014; Hinton et al., 2015) proposes to use the information in the logits of a learnt network to train the smaller student network. In early experiments, this was shown to be effective; networks much smaller than the original could be trained with small increases in error. However, modern deep architectures prove harder to compress. For example, a deep convolutional network cannot be trivially replaced by a feedforward architecture (Urban et al., 2017). Two methods have been proposed to deal with this. First, in Romero et al. (2015) the authors use a linear map between activations at intermediate points to produce an extra loss function. Second, in Zagoruyko & Komodakis (2017), the authors choose instead to match the activations after taking the mean over the channels and call this method *attention transfer*. In the context of this paper, we found attention transfer to be effective in our experiments, as described in Section 4.

## 3    Compression with Cheap Convolutions

Given a large, deep network that performs well on a given task, we are interested in compressing that network so that it uses fewer parameters. A flexible and widely applicable way to reduce the number

of parameters in a model is to replace all its convolutional layers with a cheaper alternative. Doing this replacement invariably impairs performance when this reduced network is trained directly on the data. Fortunately, we are able to demonstrate that modern distillation methods enable the cheaper model to have performance closer to the original large network.

## 3.1 Distillation

For this paper, we utilise and compare two different distillation methods for learning a smaller student network from a large, pre-trained teacher network: knowledge distillation (Ba & Caruana, 2014; Hinton et al., 2015) and attention transfer (Zagoruyko & Komodakis, 2017).

**Knowledge Distillation**   Let us denote the cross entropy of two probability vectors $\mathbf{p}$ and $\mathbf{q}$ as $\mathcal{L}_{CE}(\mathbf{p}, \mathbf{q}) = -\sum_k p_k \log q_k$. Assume we have a dataset of elements, with one such element denoted $\mathbf{x}$, where each element has a corresponding one-hot class label: denote the one-hot vector corresponding to $\mathbf{x}$ by $\mathbf{y}$. Given $\mathbf{x}$, we have a trained teacher network $\mathbf{t} = \text{teacher}(\mathbf{x})$ that outputs the corresponding logits, denoted by $\mathbf{t}$; likewise we have a student network that outputs logits $\mathbf{s} = \text{student}(\mathbf{x})$. To perform knowledge distillation we train the student network to minimise the following loss function (averaged across all data items):

$$\mathcal{L}_{KD} = (1 - \alpha)\mathcal{L}_{CE}(\mathbf{y}, \sigma(\mathbf{s})) + 2\alpha T^2 \mathcal{L}_{CE}\left(\sigma\left(\frac{\mathbf{t}}{T}\right), \sigma\left(\frac{\mathbf{s}}{T}\right)\right), \tag{1}$$

where $\sigma(.)$ is the softmax function, $T$ is a temperature parameter and $\alpha$ is a parameter controlling the ratio of the two terms. The first term is a standard cross entropy loss penalising the student network for incorrect classifications. The second term is minimised if the student network produces outputs similar to that of the teacher network. The idea being that the outputs of the teacher network contain additional, beneficial information beyond just a class prediction.

**Attention Transfer**   Consider some choice of layers $i = 1, 2, ..., N_L$ in a teacher network, and the corresponding layers in the student network. At each chosen layer $i$ of the teacher network, collect the spatial map of the activations for channel $j$ into the vector $\mathbf{a}_{ij}^t$. Let $A_i^t$ collect $\mathbf{a}_{ij}^t$ for all $j$. Likewise for the student network we correspondingly collect into $\mathbf{a}_{ij}^s$ and $A_i^s$.

Now given some choice of mapping $\mathbf{f}(A_i)$ that maps each collection of the form $A_i$ into a vector, attention transfer involves learning the student network by minimising:

$$\mathcal{L}_{AT} = \mathcal{L}_{CE}(\mathbf{y}, \sigma(\mathbf{s})) + \beta \sum_{i=1}^{N_L} \left\| \frac{\mathbf{f}(A_i^t)}{||\mathbf{f}(A_i^t)||_2} - \frac{\mathbf{f}(A_i^s)}{||\mathbf{f}(A_i^s)||_2} \right\|_2, \tag{2}$$

where $\beta$ is a hyperparameter. Zagoruyko & Komodakis (2017) recommended using $\mathbf{f}(A_i) = (1/N_{A_i}) \sum_{j=1}^{N_{A_i}} \mathbf{a}_{ij}^2$, where $N_{A_i}$ is the number of channels at layer $i$. In other words, the loss targeted the difference in the spatial map of average squared activations, where each spatial map is normalised by the overall activation norm.

Let us examine the loss (2) further. The first term is again a standard cross entropy loss. The second term, however, ensures the spatial distribution of the student and teacher activations are similar at selected layers in the network, the explanation being that both networks are then *paying attention* to the same things at those layers.

## 3.2 Cheap Convolutions

As large fully-connected layers are no longer commonplace, convolutions make up almost all of the parameters in modern networks.[1] It is therefore desirable to make them smaller. Here, we present several convolutional blocks that may be introduced in place of a standard block in a network to substantially reduce its parameter cost.

First, let us consider a standard two dimensional convolutional layer that contains $N_{\text{out}}$ filters, each of size $N_{\text{in}} \times k \times k$. $N_{\text{out}}$ is the number of channels of the layer output, $N_{\text{in}}$ is the number of

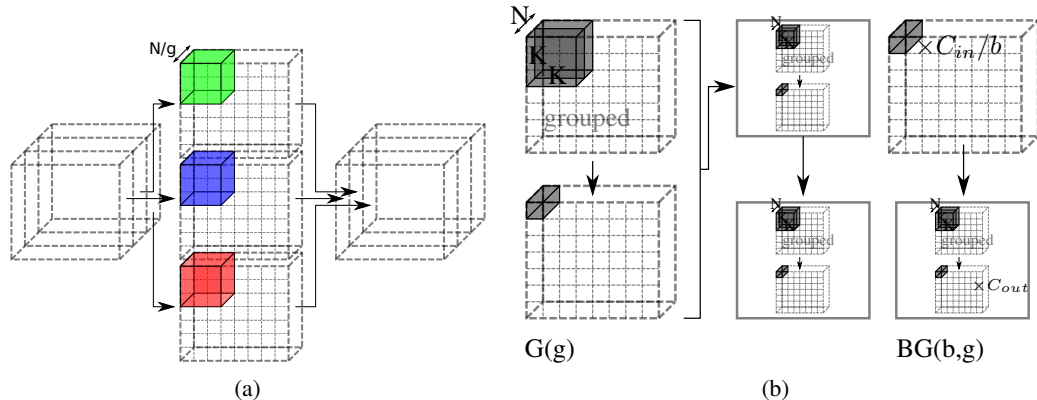

Figure 1: In (a) a grouped convolution operates by passing independent filters over the tensor after it is separated into $g$ groups over the channel dimension; as each of the $g$ filters needs only to operate over $N/g$ channels this reduces the parameter cost of the layer by a factor of $g$. These can be composed into the blocks illustrated in (b). The *Grouped + Pointwise* ($G(g)$) block substitutes a $k \times k$ convolution with a grouped convolution followed by a pointwise ($1 \times 1$) convolution, repeating this twice. To reduce parameters further, a pointwise *Bottleneck* can be used before the Grouped + Pointwise convolution ($BG(b,g)$).

channels of the input, and $k \times k$ is the kernel size of each convolution. In modern networks it is almost always the case that $N_{\text{in}} \leqslant N_{\text{out}}$. Let $N = \max(N_{\text{in}}, N_{\text{out}})$. Then the parameter cost of this layer is $N_{\text{in}} N_{\text{out}} k^2$, and is bounded by $N^2 k^2$. In a typical residual network, a block contains two such convolutions. We will refer to this as a *Standard* block $S$, and it is outlined in Table 1.

An alternative approach is to separate each convolution into $g$ groups, as shown in Figure 1a. By restricting the convolutions to only mix channels within each group, we obtain a substantial reduction in the number of parameters for a grouped computation: for example, for $N_{\text{in}} = N_{\text{out}} = N$ the cost changes from $N^2 k^2$ for a standard layer to $g$ groups of $(N/g)^2 k^2$ parameter convolutions, hence reducing the parameter cost by a factor of $g$. We can then provide some cross-group mixing by following each grouped convolution with a pointwise convolution, with a $N^2$ parameter cost (when $N_{\text{in}} \neq N_{\text{out}}$ the change in channel size occurs across this pointwise convolution). We refer to this substitution operator as $G(g)$ (grouped convolution with $g$ groups) and illustrate it in Figure 1b.

In the original ResNet paper (He et al., 2016a) the authors introduced a bottleneck block which we have parameterised, and denoted as $B(b)$ in Table 1: the input first has its channels decreased by a factor of $b$ via a pointwise convolution, before a full convolution is carried out. Finally, another pointwise convolution brings the representation back up to the desired $N_{out}$. We can reduce the parameter cost of this block even further by replacing the full convolution with a grouped one; the *Bottleneck Grouped + Pointwise* block is referred to as $BG(b,g)$ and is illustrated in Figure 1b.

These substitute blocks are compared in Table 1 and their computational costs are given. In practice, by varying the bottleneck size and the number of groups, network parameter numbers may vary over two orders of magnitude; enumerated examples are given in Table 2.

Using grouped convolutions and bottlenecks are common methods for parameter reduction when designing a network architecture. Both are easy to implement in any deep learning framework. Sparsity inducing methods (Han et al., 2016), or approximate layers (Yang et al., 2015), may also provide advantages, but these are complementary to the approaches here. More structured reductions such as grouped convolutions and bottlenecks can be advantageous over sparsity methods in that the sparsity structure does not need to be stored. In the following sections, we demonstrate that using these proposed blocks with effective model distillation allows for substantial compression with minimal reduction in performance.

## 4   CIFAR Experiments

In this section we train and evaluate a number of student networks, each distilled from the *same* large teacher network. Experiments are conducted for both the CIFAR-10 and CIFAR-100 datasets. We distil with (i) knowledge distillation and (ii) attention transfer. We also train the networks without

Table 1: Convolutional Blocks used in this paper: a standard block $S$, a grouped + pointwise block $G$, a bottleneck block $B$, and a bottleneck grouped + pointwise block $BG$. Conv refers to a $k \times k$ convolution. GConv is a grouped $k \times k$ convolution and Conv1x1 is a pointwise convolution. Blocks use pre-activations (He et al., 2016b): all convolutions are preceded by a batch-norm layer + a ReLU activation. We assume that the input and output to each block has $N$ channels and that channel size does not change over a particular convolution unless written out explicitly as $(x \rightarrow y)$. Where applicable, $g$ is the number of groups in a grouped convolution and $b$ is the bottleneck contraction. We give the cost of the convolutions in each block in terms of these parameters. The batch-norm cost at test time is also given, but is markedly smaller.

| Block | $S$ | $G(g)$ | $B(b)$ | $BG(b,g)$ |
|---|---|---|---|---|
| Structure | Conv<br>Conv | GConv (g)<br>Conv1x1<br>GConv (g)<br>Conv1x1 | Conv1x1$(N \rightarrow \frac{N}{b})$<br>Conv<br>Conv1x1$(\frac{N}{b} \rightarrow N)$ | Conv1x1$(N \rightarrow \frac{N}{b})$<br>GConv(g)<br>Conv1x1$(\frac{N}{b} \rightarrow N)$ |
| Conv Params | $2N^2k^2$ | $2N^2(\frac{k^2}{g}+1)$ | $N^2(\frac{k^2}{b^2}+\frac{2}{b})$ | $N^2(\frac{k^2}{gb^2}+\frac{2}{b})$ |
| BN Params | $4N$ | $8N$ | $N(2+\frac{4}{b})$ | $N(2+\frac{4}{b})$ |

any form of distillation (i.e. from scratch) to observe whether the distillation process is necessary to obtain good performance. In this way we demonstrate that the high performance comes from the distillation, and cannot be achieved by directly training the student networks using the data.

For comparison we also study student networks with smaller architectures (i.e. fewer layers/filters) than the teacher. This enables us to test if the block transformations we propose are key, or it is simply a matter of distilling networks with smaller numbers of parameters. We compare the smaller student architectures with student architectures implementing cheap, substitute convolutional blocks, but with the same architecture as the teacher. The different convolutional blocks are summarised in Table 1 and the student networks are described in detail in Section 4.1. Results are given in Table 2 and Figure 2. These results are discussed in detail in Section 4.2.

## 4.1 Network Descriptions

For our experiments we utilise the Wide Residual Network (WRN) architecture (Zagoruyko & Komodakis, 2016); the bulk of the network lies in its $\{conv2, conv3, conv4\}$ groups and the network depth $d$ determines the number of convolutional blocks $n$ in these groups as $n = (d-4)/6$. The network width, denoted by $k$, affects the channel size of the filters in these blocks. Note that when we employ attention transfer the student and teacher outputs of groups $\{conv2, conv3, conv4\}$ are used as $\{A_1, A_2, A_3\}$ in the second term of Equation (2) with $N_L = 3$.

For our teacher network we use WRN-40-2 (a WRN with depth 40 and width multiplier 2) with standard ($S$) blocks. $3 \times 3$ kernels are used for all non-pointwise convolutions in our student and teacher networks unless stated otherwise. For our student networks we use:

- WRN-40-1, 16-2, and 16-1 with $S$ blocks. These are student networks that are thinner and/or more shallow than the teacher and represent typical student networks used.

- WRN-40-2 with $S$ blocks where the $3 \times 3$ kernels have been replaced with $2 \times 2$ dilated kernels (as described in Yu & Koltun (2016)). This allows us to see if it possible to naively reduce parameters by effectively zeroing out elements of standard kernel.

- WRN-40-2 using a bottleneck block $B$ with $2\times$ and $4\times$ channel contraction ($b$).

- WRN-40-2 using a grouped + pointwise block $G$ for group sizes ($g$) {2, 4, 8, 16, N/16, N/8, N/4, N/2, N} where N is the number of channels in a given block. This allows us to explore the spectrum between full convolutions ($g = 1$) and fully separable convolutions ($g = N$).

- WRN-40-2 with a bottleneck grouped + pointwise block $BG$. We use $b = 2$ with groups sizes of {2, 4, 8, 16, M/16, M/8, M/4, M/2, M} where $M = N/b$ is the number of channels *after the bottleneck*. We use this notation so that $g = M$ represents fully separable convolutions and we can easily denote divisions thereof. $BG(4, M)$ is also used to observe the effect of extreme compression.

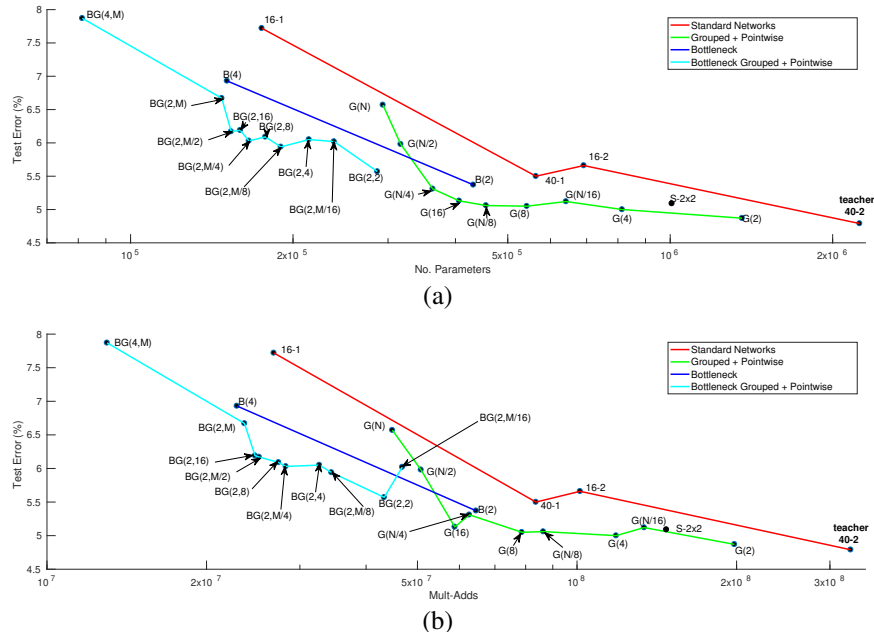

Figure 2: Test Error vs. (a) No. parameters and (b) Mult-adds for student networks learnt with attention transfer on CIFAR-10. Note that the x-axes are log-scaled. Points on the red curve correspond to networks with $S$ convolutional blocks and reduced architectures. All other networks have the same WRN-40-2 architecture as the teacher but with cheap convolutional blocks: $G$ (green), $B$ (blue), and $BG$ (cyan). The blocks are described in Table 1. Notice that the student networks with cheap blocks outperform those with smaller architectures and standard convolutions for a given parameter budget or mult-add budget.

**Implementation Details** For training we used minibatches of size 128. Before each minibatch, the images were padded by $4 \times 4$ zeros, and then a random $32 \times 32$ crop was taken. Each image was left-right flipped with a probability of a half. Networks were trained for 200 epochs using SGD with momentum fixed at 0.9 with an initial learning rate of 0.1. The learning rate was reduced by a factor of 0.2 at the start of epochs 60, 120, and 160. For knowledge distillation we set $\alpha$ to 0.9 and used a temperature of 4. For attention transfer $\beta$ was set to 1000. The code to reproduce these experiments is available at `https://github.com/BayesWatch/pytorch-moonshine`.

### 4.2 Analysis and Observations

Figure 2a compares the parameter cost of each student network (on a log scale) against the test error on CIFAR-10 obtained with attention transfer. On this plot, the ideal network would lie in the bottom-left corner (few parameters, low error). What is fascinating is that almost every network with the same architecture as the teacher, but with cheap convolutional blocks (those on the blue, green, and cyan lines) performs better for a given parameter budget than the reduced architecture networks with standard blocks (the red line). $BG(2, 2)$ outperforms 16-2 (5.57% vs. 5.66%) despite having considerably fewer parameters (287K vs. 692K). Several of the networks with $BG$ blocks both significantly outperform 16-1 and use fewer parameters.

It is encouraging that significant compression is possible with only small losses; several networks perform almost as well as the teacher with considerably fewer parameters – $G(N/8)$ has an error of 5.06%, close to that of the teacher, but has just over a fifth of the parameters. $BG(2, M/8)$ has less than a tenth of the parameters of the teacher, for a cost of 1.15% increase in error. Even simply switching all convolutions with smaller, dilated equivalents ($S - 2 \times 2$) allows one to use half the parameters for a similar performance.

An important lesson can be learnt regarding grouped + pointwise convolutions. They are often used in their depthwise-separable (Chollet, 2016) form ($g = N$). However, the networks with half, or quarter that number of groups perform substantially better for a modest increase in parameters. $G(N/4)$ has

363K parameters compared to the 294K of $G(N)$ but has an error that is $1.26\%$ lower. The number of groups is an easy parameter to tune to trade some performance for a smaller network. Grouped + pointwise convolutions also work well in conjunction with a bottleneck of size 2, although for large bottlenecks the error increases significantly, as can be seen for $BG(4, M)$. Despite this, it is still of comparable performance to 16-1 with half the parameters. Similar trends are observed for CIFAR-100 in Table 2b.

We also observe that training a student with attention transfer is substantially better than using knowledge distillation, or simply training from scratch. Consider Table 2, which shows the attention transfer errors of Figure 2 (the AT column) alongside those of networks trained with knowledge distillation (KD), and no distillation i.e. from scratch (Scr) for CIFAR-10 and CIFAR-100. In all cases, the student network trained with attention transfer is better than the student network trained by itself – the distillation process appears to be necessary. Some performances are particularly impressive; on CIFAR-10, for $G(2)$ blocks the error is only $0.08\%$ higher than the teacher despite the network having 60% of the parameters.

These results support our claim that greater model compression through distillation is possible by substituting the convolutional blocks in a network, rather than by shrinking its architecture. We have also demonstrated that the blocks outlined in Table 1 are suitable substitutes. By observing Figure 2b we can also see that our networks with cheap, substitute blocks utilise fewer mult-add operations than their standard equivalents, which roughly corresponds to a faster runtime. However, it is worth noting that actual runtime on a given platform or device is dependent on specifics (memory paging, choice of libraries etc.), so mult-adds are not always fully indicative of runtime, but are a decent approximation in a platform/implementation-agnostic setting.

## 5 Additional Experiments

Section 4 demonstrates the effectiveness of *cheapening* convolutions for CIFAR classification. In this section, we apply this method to two further problems. Firstly, in Section 5.1 we examine whether the benefits observed hold for large-scale image classification on ImageNet (Russakovsky et al., 2015) where there are far more classes (1000), and the images are significantly larger. Secondly, in Section 5.2 we cheapen the convolutions of a network trained for semantic segmentation.

### 5.1 ImageNet

Our experiments use a pre-trained ResNet-34 (He et al., 2016a) (21.8M parameters) as a teacher and we train several networks using attention transfer (AT). We compare student networks that have the architecture of ResNet-34, with *cheaper* convolutions, to those that have reduced architectures, and full convolutions. Note, that the bulk of the parameters in a ResNet are contained in four groups, as opposed to the three of a WideResNet. We train the following student networks: (i) ResNet-18, (ii) ResNet-18 with the channel widths of the last three groups halved (Res18-0.5), ResNet-34 with each convolutional block replaced by (iii) a $G(N)$ block[2] and (iv) a $G(4)$ block. Validation errors for these networks are available in Table 3.

Consider Res34-G(N) and Res18-0.5, which both have roughly the same parameter cost ($\sim$3M). After distillation, the former has a significantly lower top-5 error ($10.66\%$ vs. $15.02\%$). This again supports our claim that is is preferable to cheapen convolutions, rather than shrink the network architecture. Res34-G(N) trained from scratch has a noticeably higher top-5 error ($12.26\%$), it benefits from distillation. Conversely, distillation makes Res18-0.5 slightly worse, suggesting that it has no further representational capacity.

Res34-G(4) similarly outperforms Res18 (these are roughly similar in cost at 8.1M and 11.7M parameters respectively), although in this case the latter does benefit from distillation. It is intriguing that Res34-G(4) trained from scratch is actually on par with the original teacher (having a 0.12% lower top-1 error, and a 0.05% higher top-5 error) despite having 13 million fewer parameters; this generalisation capability of grouped convolutions in networks has been observed previously by Ioannou et al. (2017). Distillation is able to push its performance slightly further to the point that its top-5 error surpasses that of the teacher ($8.43\%$ vs. $8.57\%$).

Table 2: Student Network test error on CIFAR-10/100. Each network is a WideResNet with its depth-width (D-W) given in the first column, and with its block type (corresponding to Table 1 in the second. $N$ refers to the channel width of each block, and $M$ refers to the channel width after the bottleneck where applicable. The total parameter cost of the networks for CIFAR-10 is given, as well as the number of mult-add operations they use. Note that CIFAR-100 networks use an extra 11.6K parameters and mult-adds over their CIFAR-10 equivalents as they have a larger linear classification layer. Errors are reported for (i) learning with no distillation i.e. from scratch (Scr), (ii) knowledge distillation with a teacher (KD), and attention transfer with a teacher (AT). The same teacher is used for training, and is given in the first row. This table shows that (i) through attention transfer it is possible to cut the number of parameters of a network, but retain high performance and (ii) for a similar number of parameters, students with cheap convolutional blocks outperform those with expensive convolutions and smaller architectures.

| D-W | Block | Params (K) | MAdds (M) | CIFAR-10 Scr | KD | AT | CIFAR-100 Scr | KD | AT |
|---|---|---|---|---|---|---|---|---|---|
| **T** 40-2 | S | 2243.5 | 328.3 | 4.79 | – | – | 23.85 | – | – |
| 16-2 | S | 691.7 | 101.4 | 6.53 | 6.03 | 5.66 | 27.63 | 27.97 | 27.24 |
| 40-1 | S | 563.9 | 83.6 | 6.48 | 6.39 | 5.50 | 29.64 | 30.21 | 28.24 |
| 16-1 | S | 175.1 | 26.8 | 8.81 | 8.75 | 7.72 | 34.00 | 37.28 | 33.74 |
| 40-2 | S-2x2 | 1007.1 | 147.4 | 5.89 | 6.03 | 5.09 | 27.20 | 26.98 | 26.09 |
| 40-2 | G(2) | 1359.0 | 198.1 | 5.30 | 5.37 | 4.87 | 25.94 | 24.92 | 24.45 |
| 40-2 | G(4) | 814.7 | 118.5 | 5.50 | 5.81 | 5.00 | 26.20 | 25.48 | 25.30 |
| 40-2 | G(8) | 542.5 | 78.7 | 5.92 | 5.72 | 5.05 | 26.49 | 26.64 | 25.71 |
| 40-2 | G(16) | 406.4 | 58.8 | 6.65 | 6.38 | 5.13 | 28.85 | 27.10 | 26.34 |
| 40-2 | G(N/16) | 641.3 | 133.9 | 5.72 | 5.72 | 5.12 | 27.08 | 26.11 | 25.78 |
| 40-2 | G(N/8) | 455.8 | 86.4 | 6.07 | 5.61 | 5.06 | 27.85 | 27.05 | 26.15 |
| 40-2 | G(N/4) | 363.1 | 62.6 | 6.93 | 6.45 | 5.31 | 28.91 | 27.93 | 26.85 |
| 40-2 | G(N/2) | 316.7 | 50.8 | 7.12 | 6.83 | 5.98 | 30.24 | 28.89 | 28.54 |
| 40-2 | G(N) | 293.5 | 44.8 | 8.51 | 8.01 | 6.57 | 31.84 | 29.99 | 30.06 |
| 40-2 | B(2) | 431.8 | 64.5 | 6.36 | 6.28 | 5.37 | 28.27 | 28.08 | 26.68 |
| 40-2 | B(4) | 150.9 | 22.8 | 7.94 | 7.83 | 6.93 | 31.63 | 33.63 | 30.56 |
| 40-2 | BG(2,2) | 286.7 | 43.3 | 6.12 | 6.25 | 5.57 | 28.51 | 28.82 | 28.28 |
| 40-2 | BG(2,4) | 214.1 | 32.7 | 6.75 | 6.75 | 6.05 | 29.39 | 29.25 | 28.54 |
| 40-2 | BG(2,8) | 177.8 | 27.3 | 6.94 | 6.98 | 6.09 | 30.21 | 29.34 | 28.89 |
| 40-2 | BG(2,16) | 159.7 | 24.7 | 6.77 | 6.97 | 6.19 | 30.57 | 30.54 | 29.46 |
| 40-2 | BG(2,M/16) | 238.3 | 46.8 | 6.26 | 6.50 | 6.02 | 29.69 | 28.69 | 29.05 |
| 40-2 | BG(2,M/8) | 189.9 | 34.4 | 6.75 | 6.49 | 5.94 | 29.09 | 29.13 | 28.16 |
| 40-2 | BG(2,M/4) | 165.7 | 28.2 | 7.06 | 7.15 | 6.03 | 30.42 | 30.28 | 28.60 |
| 40-2 | BG(2,M/2) | 153.6 | 25.1 | 7.45 | 7.47 | 6.17 | 30.44 | 30.66 | 29.51 |
| 40-2 | BG(2,M) | 147.6 | 23.6 | 7.95 | 7.99 | 6.67 | 30.90 | 31.18 | 30.03 |
| 40-2 | BG(4,M) | 81.4 | 13.0 | 9.04 | 8.61 | 7.87 | 33.64 | 37.34 | 32.89 |

**Implementation Details**  Models were trained for 100 epochs using SGD with an initial learning rate of 0.1, momentum of 0.9 and weight decay of $10^{-4}$. The learning rate was reduced by a factor of 10 every 30 epochs. Minibatches of size 256 were used across 4 GPUs. When trained with a teacher, an additional AT loss was used with the outputs of the four groups of each ResNet. $\beta$ was set to 750 so that the total contribution of the AT loss was the same as in Section 4.

## 5.2 Semantic Segmentation

We have shown that cheapening the convolutions of a network, coupled with a good distillation process, has allowed for a substantial reduction in the number of network parameters in return for a small drop in performance. However, the networks trained thus far have all had the same task – image classification. Here, we take an existing network, trained for the task of semantic segmentation and apply our method to distil it.

For our teacher network we use an ERFNet (Romera et al., 2017a,b) that has been trained *from scratch* on the Cityscapes dataset (Cordts et al., 2016) – a collection of images of urban street scenes, in which each pixel has been labelled as one of 19 classes. The bulk of an ERFNet is made up of standard residual blocks where each full convolution has been replaced by a pair of 1D alternatives:

Table 3: Top 1 and Top 5 classification errors (%) on the validation set of ImageNet for models (i) trained from scratch, and (ii) those trained with attention transfer with ResNet-34 (Res34) as a teacher. Res18 refers to a ResNet-18, and Res18-0.5 is a ResNet-18 where the channel width in the last three groups is halved. Res34-G($x$) is a ResNet-34 with each convolutional block replaced by a G($x$) block. We can observe that for a particular parameter budget (3M or $\sim$10M) the networks with cheap replacement blocks outperform those with reduced architectures. These trends follow for mult-adds. Note that the Res34 and Res18 scratch results were obtained from pre-trained PyTorch models.

| Model | Params | Mult-Adds | Scratch Top 1 | Scratch Top 5 | AT Top 1 | AT Top 5 |
|---|---|---|---|---|---|---|
| Res34 **T** | 21.8M | 3.669G | 26.73 | 8.57 | – | – |
| Res18 | 11.7M | 1.818G | 30.36 | 11.02 | 29.18 | 10.05 |
| Res34-G(4) | 8.1M | 1.395G | 26.61 | 8.62 | 26.58 | 8.43 |
| Res18-0.5 | 3.2M | 909M | 36.96 | 15.01 | 37.20 | 15.02 |
| Res34-G(N) | 3.1M | 559M | 32.98 | 12.26 | 30.16 | 10.66 |

Table 4: IoU accuracy (%) on the validation set of Cityscapes for (i) ERFNet, and (ii) ERFNet with replacement blocks (ERFNet-G(N)). For ERFNet-G(N) the accuracy when trained from scratch (Scratch IoU), and when used as a student with the original ERFNet as a teacher (AT IoU) is given.

| Model | Params | Mult-Adds | Scratch IoU | AT IoU |
|---|---|---|---|---|
| ERFNet | 2.06M | 3.73G | 70.59 | – |
| ERFNet-G(N) | 0.49M | 1.19G | 65.29 | 68.11 |

a $3 \times 1$ convolution followed by a $1 \times 3$ convolution. The second such pair in each block is often dilated. To *cheapen* this network for use as a student, we replace each block with a $G(N)$ block, maintaining the dilations where appropriate.

We use the same optimiser, and training schedule as for the original ERFNet. When training the student the only difference is the addition of an attention transfer term (see Equation 2) between several of the feature maps in the final loss. The models are evaluated using class Intersection-over-Union (IoU) accuracy on the validation set, and the results can be found in Table 4.

In Romera et al. (2017b) the authors detail how ERFNet is designed with efficiency in mind. With only one training run and no tuning, we are able to reduce the number of parameters to one quarter of the original for a modest drop in performance.

**Implementation Details** Models were trained using the same optimiser, schedule, and image scaling and augmentation as in the ERFNet paper (Romera et al., 2017b) with the attention transfer loss for the case of ERFNet-G(N) as a student. For encoder training, the outputs of layers 7, 12, and 16 were used for attention transfer with $\beta = 1000$. For decoder training, the outputs of layers 19 and 22 were also used and $\beta$ was dropped to 600 (so that the contribution of this term remains this same).

## 6 Conclusion

After training a large, deep model it may be prohibitively time consuming to design a model compression strategy in order to deploy it. On many problems, it may also be more difficult to achieve the desired performance with a smaller model. We have demonstrated a model compression strategy that is fast to apply, and doesn't require any additional engineering for both image classification and semantic segmentation. Furthermore, the optimisation algorithm of the larger model is sufficient to train the cheaper student model.

The cheap convolutions used in this paper were chosen for their ease of implementation. Future work could investigate more complicated approximate operations, such as those described in Moczulski et al. (2016); which could make a difference for the $1 \times 1$ convolutions in the final layers of a network. One could also make use of custom blocks generated through a large scale black box optimisation as in Zoph et al. (2018). Equally, there are many methods for low rank approximations that could be applicable (Sainath et al., 2013; Jaderberg et al., 2014; Garipov et al., 2016). We hope that this work encourages others to consider *cheapening their convolutions* as a compression strategy.

**Acknowledgements.**    This project has received funding from the European Union's Horizon 2020 research and innovation programme under grant agreement No. 732204 (Bonseyes). This work is supported by the Swiss State Secretariat for Education, Research and Innovation (SERI) under contract number 16.0159. The opinions expressed and arguments employed herein do not necessarily reflect the official views of these funding bodies. The authors are grateful to Sam Albanie, Luke Darlow, Jack Turner, and the anonymous reviewers for their helpful suggestions.

## Footnotes

[1]The parameters introduced by batch normalisation are negligible compared to those in the convolutions. However, they are included for completeness in Table 1.

[2]As the convolutional blocks in the teacher do not use pre-activations, the $G$ blocks used here are modified accordingly (BN + ReLU now come after each convolution). This also applies to the networks in Section 5.2.

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
