[Reviews · NeurIPS 2018]

Reviewer 1



The paper explores the tradeoffs involved when selecting a student architecture for models trained with distillation. While much of the research so far has focused on simpler architectures (fewer layers, less depth in each layer), this paper proposes keeping the depth and number of layers of a state-of-the-art architecture and instead achieving parameter reduction by replacing convolutional blocks with simpler convolutional blocks. Three techniques present in previous research are proposed to simplify the convolutional blocks: grouping (where the channels are divided in groups, each group has a convolution of the desired depth applied to it, then a 1x1 convolution is used to mix the channels), bottlenecking (where a convolution with N channels is replaced with three convolutions; first a 1x1 convolution to reduce the depth; then a full convolution; finally a 1x1 convolution to increase back the depth), and a combination of grouping and bottlenecking. Interestingly it seems like the different simplified convolutional blocks can lead to a better tradeoff between number of parameters / computation and accuracy, when distilling from a state-of-the-art teacher model, than reducing the number of layers and depth. The evaluation could have been a little more thorough as it doesn't look like the hyperparameters were retuned for each architecture, which might hide some potential improvements. That said the idea of replacing convolutions with simpler convolutions when building student models for distillation is interesting, and should be further investigated.

Reviewer 2



Main idea: this paper proposes to tackle the task of speeding up any CNN by replacing the "higher level blocks" that typically consist of stacked convolutions with cheaper blocks consisting of 1x1 convolutions and potentially group convolutions. Thus far this is not super-novel, as this idea has been studied before. However, the authors also employ knowledge distillation (again this idea has been studied before in isolation) to match the inner layer block activations between the teacher and the student (cheaper) networks. This shows very promising results over the baseline which only replaces the blocks with their cheaper version without distillation. The main selling point of this paper is that the ideas have been proven before independently, and this paper ties them together nicely. Therefore, it should be relatively easy for other researchers to hopefully replicate these results. Another nice plus is that the authors included a segmentation task besides classification. This is important because in the literature most of the work done in speeding up networks seems to be centered around classification, and it's not clear whether such methods generalize beyond this niche domain. In fact, from personal experience, we have also experienced a huge quality degradation when changing the convolutions to be depthwise, but not doing distillation in the context of non-classification problems. On the negative side, I would have wished to see a discussion about runtime. Most of the focus of the paper is in comparing the number of parameters against the various errors. For this to be truly useful in practice it would be great to discuss how does the number of FLOPs (or forward pass time) change (did it increase?, did it decrease?). I would strongly suggest the authors clarify this in the final version of the paper. Overall score: I think the paper has some good practical applications, and I think we should accept it. Confidence: I am not super familiar with the field of making networks faster besides the ideas I mentioned above, and I did do a bit of research on Google Scholar trying to find similar papers that discuss both ideas, but I didn't find anything matching quite exactly (e.g., MobileNets seem to use both distillation and cheaper convolutions but they don't match the internal features).

Reviewer 3



This paper provides an extensive experimental study showing that one is able to distill the knowledge of large networks using much smaller networks, and notably the paper highlights the fact via the use of "cheap convolutions" - group convolutions and 1x1 convolutions as proposed in the papers in the recent years. I find the experimentation part of the paper to be sufficient, and I believe it would be very useful data points for the community. However, I am not very certain about the novelty of the paper; it summarizes the two reasonably known approaches (knowledge distillation and attention transfer), and utilizes them in the specific case of training cheap convolutions. The convolution types are known for a while too, for example the 1x1 convolution, first appearing in the network-in-network paper, and group convolution which was first in AlexNet and recently seeing more interest since ResNet. As a result, this paper seems to be more on experiment verification instead of proposing novel approaches. I am personally fine with the theme of the paper, but do feel that when comparing to other novel papers, it might fall short of the originality axis a little bit. As a result I would like to recommend a marginal acceptance.

Reviewer 4



The paper under review examines the problem of designing smaller CNNs for lightweight devices. The approach advocated by the paper is to start from a given (heavyweight) teacher architecture and replace its building blocks with faster ones. Most experiments in the paper use a wide ResNet as the reference heavyweight architecture and replace the standard convolutional layers in the ResNet blocks with grouped/bottleneck CNN layers which use fewer weights and lend themselves to faster inference. For training the resulting lightweight architecture, the paper advocates the use of distillation, where the training loss contains both the standard classification loss coming from training data and knowledge distillation or attention transfer from the more accurate teacher network. The paper reports results mainly on the CIFAR-10 and CIFAR-100 datasets, as well as more limited experiments on the Imagenet classification task and Cityscapes semantic segmentation task. The paper is well written and I find the proposed approach well motivated. However, a major shortcoming of the paper in its current version is that it only compares the different architectures in terms of number of parameters (which determines the memory footprint). The other major axis to compare different CNNs is in terms of number of operations (MAdds) which is more correlated to the runtime. This is unfortunate, as the corresponding MAdd numbers are very easy to compute for a given input image size, and does not allow direct runtime comparison of the resulting networks to other recent papers that propose alternative small network designs.